# Association between Heavy Metals, Bisphenol A, Volatile Organic Compounds and Phthalates and Metabolic Syndrome

**DOI:** 10.3390/ijerph16040671

**Published:** 2019-02-25

**Authors:** Yun Hwa Shim, Jung Won Ock, Yoon-Ji Kim, Youngki Kim, Se Yeong Kim, Dongmug Kang

**Affiliations:** 1Department of Premedicine, School of Medicine, Pusan National University, Yangsan, Gyeongnam 50612, Korea; syh9832@pusan.ac.kr (Y.H.S.); jungwon1292@pusan.ac.kr (J.W.O.); 2Department of Preventive, and Occupational & Environmental Medicine, School of Medicine, Pusan National University, Yangsan, Gyeongnam 50612, Korea; harrypotter79@pusan.ac.kr (Y.-J.K.); mungis@pusan.ac.kr (Y.K.); 3Department of Occupational and Environmental Medicine, Pusan National University Yangsan Hospital, Yangsan, Gyeongnam 50612, Korea; 30white@pusan.ac.kr

**Keywords:** metabolic syndrome, volatile organic compound, phthalate, muconic acid, MEHHP

## Abstract

The incidence of metabolic syndrome (MetS), which causes heart disease and stroke, has increased significantly worldwide. Although many studies have revealed the relationship between heavy metals (cadmium, mercury, and lead), the sum of metabolites of di(2-ethylhexyl) phthalate (DEHP), and MetS, the results remain inconsistent. No study has reported the association between various volatile organic compounds (VOCs) and phthalate metabolites with MetS. This cross-sectional study of a representative sample of adult South Koreans aimed to evaluate the relationship between heavy metals, VOC metabolites, phthalate metabolites, bisphenol A and MetS after adjusting for demographic variables. Data from the Korean National Environmental Health Survey II (2012–2014) (*n* = 5251) were used in the analysis. Multiple logistic regression analysis was performed for MetS with log-transformed hazardous material quartiles after covariate adjustment. Urine muconic acid (MuA) and mono- (2-ethyl-5-hydroxyhexyl) phthalate (MEHHP) levels were significantly associated with MetS after adjusting for confounders (odds ratio: 1.34 and 1.39, respectively). Urine MuA and MEHHP levels were significantly associated with MetS. Because of the rarity of this study, which investigated the relationship between each VOC and phthalate metabolite with MetS and the strict definition of all indirect measures of MetS components, further research is needed.

## 1. Introduction

Metabolic syndrome (MetS) is defined as a combination of central obesity, increased fasting triglyceride levels, reduced high-density lipoprotein (HDL) cholesterol, hypertension, and impaired fasting glucose [1]. The incidence of MetS is high worldwide. In the United States, 33% of adults have MetS [2]. In the Asia-Pacific region, nearly one-fifth of the adult population or more were affected by MetS and its prevalence is gradually increasing [3]. MetS is an important risk factor for cardiovascular disease incidence and mortality [4] and chronic kidney disease progression [5,6]. Moreover, a previous study suggested that MetS is associated with an increased risk of breast cancer [7,8] and Parkinson’s disease [9]. MetS is a comprehensive condition caused by genetic predisposition and environmental factors. Smoking [10], alcohol consumption, and lack of physical activity are well-known lifestyle factors associated with MetS.

Exposures to environmental chemicals are associated with the development of MetS [11]. Many studies [12,13,14] revealed the relationship between heavy metals (cadmium, mercury, and lead) and MetS, but the results were not consistent. In a study using data from the 2008 Korean National Health and Nutrition Examination Survey (KNHANES), only blood lead levels were significantly associated with MetS [15]. In another Korean study, no significant results were observed among women. In contrast, men with the highest tertile of blood cadmium were more likely to have MetS than those with the lowest tertile [16]. Other studies suggested the association between blood mercury concentration and MetS [17,18]. In the case of bisphenol A (BPA), a few studies [19,20] have reported that there is no association between urine BPA levels and diabetes, while other studies [21,22,23] showed different results. Many studies have reported the relationship between BPA and MetS (or MetS components; diabetes), but the results were inconsistent. Various studies have investigated the association of phthalate exposure with obesity [24], insulin resistance [25,26], and high blood pressure [27]. However, little is known about the association of phthalate exposure and MetS, and the results were inconsistent [28,29]. Representative phthalate metabolites include mono(2-ethyl-5-hydroxyhexyl) phthalate (MEHHP), mono(2-ethyl-5-oxohexyl) phthalate (MEOHP), and mono (2-ethyl-5-carboxypentyl) phthalate (MECPP), which are metabolites of di(2-ethylhexyl) phthalate (DEHP); mono-*n*-butyl phthalate (MnBP), which is a metabolite of di-*n*-butyl phthalate; and monobenzyl phthalate (MBzP), which is a metabolite of benzyl butyl phthalate. One study showed the association between MnBP and MetS in adolescents [28]. However, another study revealed that higher DEHP metabolite concentration (the sum of MEHP, MEHHP, and MEOHP) was associated with an increased odds of MetS in men [29]. Although studies on volatile organic compounds (VOCs), which are important environmental chemicals including toluene, benzene, ethylbenzene and styrene, styrene, and xylene, have been actively studied in relation to air pollution [30], lung cancer [31], and childhood acute lymphocytic leukemia [32], a few studies have reported the relationship between any VOCs and MetS. A study [33] on the cardiovascular disease risk of benzene (a kind of VOC) showed that there is a significant relationship between benzene and hypertension and hyperlipidemia, which are components of MetS. 

We aimed to conduct a study to evaluate the relationship between MetS and various environmental hazardous materials including various heavy metals, BPA, phthalates, and VOCs.

## 2. Materials and Methods 

### 2.1. Study Participants and Demographic Variables

This study was based on the Korean National Environmental Health Survey II (2012–2014, KNEHS). KNEHS is conducted annually using a rolling sampling design that involves a complex, stratified, multistage, probability-cluster survey of a representative sample of the civilian population of South Korea. The survey consisted of three components: a health interview survey, a health examination survey, and an environmental pollutant examination survey. All participants provided a written consent and the study was approved by the internal review board (National Institute of Environmental Researches, Department of Environmental Health Research—1805) (IRB NIER 1805). This survey was conducted among adults aged over 20 years. We excluded the participants whose blood lead and mercury, urinary cadmium, bisphenol A, phthalates’ metabolite, and VOCs’ metabolite levels were not measured (*n* = 507) as well as those study variables with missing data (urinary creatinine; *n* = 709, alanine aminotransferase (ALT), aspartate aminotransferase (AST); *n* = 1). Of the 6468 participants, 5251 were included in the analysis. Data on age, education, smoking history, alcohol intake, regular exercise, income level, and marital status were collected during the health interview. Height and weight were measured, with the participants wearing light clothing and no shoes. Age, as reported at the time of the health interview, was categorized into six groups. Education level (less than high school, high school, and college or higher), smoking status (current smoker, past smoker, and never-smoker), alcohol consumption status (non-drinker, past drinker, and current drinker), regular exercise (regular or not), income level (good, average, and bad), and marital status (single, married, and divorced/separated) were categorized according to the participants’ interview responses.

### 2.2. Environmental Hazardous Material Concentrations

A total of 14 environmental hazardous materials were selected based on their known effects on diabetes, obesity, hypertension, or other cardiovascular diseases, which were closely related to MetS, and emerging environmental health issues. Concentrations of the following materials were measured: lead and mercury in the blood and cadmium in the urine; BPA in urine; MEHHP, MEOHP, MECCP, MnBP, and MBzP in the urine; and hippuric acid (HA), *trans,trans*-muconic acid (MuA), phenylglyoxylic acid (PGA), mandelic acid (MaA), and methylhippuric acid (MHA) in urine, which are metabolites of toluene, benzene, ethylbenzene and styrene, styrene, and xylene, respectively. Concentrations of blood lead and urine cadmium were analyzed by graphite furnace atomic absorption spectrometry, and gold-amalgam mercury analyzer was used for measurement of blood mercury concentration. Concentrations of phthalate metabolites, BPA, and HA were analyzed by ultra-performance liquid chromatography-tandem mass spectrometry [34], and VOC metabolites except HA were analyzed by high-performance liquid chromatography-tandem mass spectrometry [35]. Details of the instrumental analysis were described in the National Institute of Environmental Research. Concentrations of environmental hazardous materials in urine samples were adjusted for creatinine. For all environmental hazardous materials (in blood and urine), natural logarithm was taken. 

### 2.3. Definition of MetS

We defined MetS using criteria from the US National Cholesterol Education Program-Adult Treatment Panel III (NCEP ATP III) [36]. An individual was considered to have MetS if they have 3 of the following 5 components: central obesity (waist circumference >102 cm for males and >88 cm for females); a fasting plasma triglycerides level (>150 mg/dL or on specific (systolic blood pressure (BP) > 130 mm Hg or diastolic BP >85 mm Hg) or taking medications); Low HDL cholesterol (males, <40 mg/dL); females, <50 mg/dL); hypertension antihypertensive medication; and impaired fasting plasma glucose (fasting glucose >110 mg/dL) or taking medication or previously diagnosed with type-2 diabetes mellitus [1]. As BP, waist circumference, and fasting glucose levels were not measured in the KNEHS, we used an operational definition for MetS based on the following criteria: current BP medication use, obesity (body mass index (BMI) ≥ 30) [36], and current anti-diabetic medication use.

### 2.4. Statistical Analysis

IBM SPSS Statistics 23.0 (IBM, Armonk, NY, USA) was used for statistical analyses and data management. Independent sample t-test and chi-square test were used for univariate analysis to determine the relationship between variables and MetS status. Covariates that were significant in the univariate analysis were adjusted by multiple logistic regression analysis with Mets status as a dependent variable. Although smoking had no significant relationship with MetS in this study, it was adjusted because smoking was found to be related with both MetS and hazardous materials [10,37]. The concentrations of hazardous materials were divided into quartiles, and the lowest quartile subgroup was used as the reference. We conducted simple and multiple logistic regression analysis adjusting the covariates of sex, age, smoking status, drinking status, education level, marital status, and income level. *p*-values of <0.05 were considered significant. 

## 3. Results

### 3.1. Demographic Characteristics of Participants According to MetS Status

Significant differences in demographic characteristics according to MetS status were observed (Table 1). 

The MetS group had higher proportions of women (*p* = 0.018), participants with older age (*p* < 0.001), participants with obesity (*p* < 0.001), participants with low educational status (*p* < 0.001), participants with low income status (*p* < 0.001), and participants with divorced/separated status (*p* < 0.001) than the non-MetS group. The proportion of current drinkers (*p* < 0.001) was significantly higher in the non-MetS group than in the MetS group. Smoking (*p* = 0.100) and regular exercise (*p* = 0.929) were not significantly different between MetS group and non-MetS group. (Table 1).

### 3.2. Differences in Log-Transformed Blood and Urine Hazardous Material Concentrations by MetS Status

Hazardous material concentrations in participants were compared based on MetS status (Table 2). The results of the independent sample t-test showed that the concentrations of heavy metals including blood lead (*p* = 0.031) and creatinine-adjusted urine cadmium (*p* < 0.001); phthalate metabolites including MEHHP (*p* < 0.001), MEOHP (*p* < 0.001), MECCP (*p* < 0.001), MnBP (*p* < 0.001), and MBzP (*p* < 0.001); and VOC metabolites including creatinine-adjusted MuA (*p* = 0.021), PGA (*p* = 0.001), and MaA (*p* = 0.001) were significantly higher in participants with MetS than in those without MetS. However, blood mercury (*p* = 0.594), creatinine-adjusted urine HA (*p* = 0.321), and MHA (*p* = 0.077) concentrations did not show any significant difference. Additionally, BPA (*p* < 0.001) concentration was significantly lower in participants with MetS than in those without MetS (Table 2).

### 3.3. Multiple Logistic Regression Analysis between MetS Status and Environmental Hazardous Material Concentrations 

We evaluated the odds ratios (ORs) and 95% confidence intervals (CI) for MetS status and compared them with those for hazardous materials (Table 3). In the simple logistic regression analysis, the ORs for MetS associated with heavy metals including cadmium (all) and lead (third quartile); VOCs including MuA (third quartile), PGA (second and third quartiles), and MaA (second and third quartiles); phthalates including MEHHP (all), MEOHP (all), MECPP (all), MnBP (third and fourth quartiles), and MBzP (all) were significantly higher than those in the lowest quartile. In the multiple logistic regression analysis, the ORs for MetS associated with VOC metabolites including MuA (third and fourth quartiles), PGA (second quartile), and MEHHP (fourth quartile) were also higher than those in the lowest quartile. In the multiple regression analysis, the ORs of muconic acid increased in higher quartiles. Lead, cadmium, MaA, MEOHP, MECPP, MnBP, and MBzP, which appeared to be significant in the simple regression analysis, were no longer significant after adjustment of demographic variables (Table 3).

## 4. Discussion

There was a significant relationship between urinary MuA and MetS. MuA, measured in urine, is one of the metabolites of benzene. Along with urinary s-phenylmercapturic acid (s-PMA), MuA is useful for evaluating benzene exposure at low concentrations (between 0.1 ppm and 10 ppm) [38]. At low exposure levels, benzene metabolism in the liver would produce several metabolites: 70%–85% is metabolized to phenol, 5%–10% is metabolized to catechol and MuA, while less than 1% is metabolized to s-PMA [39]. Outdoor or indoor air contains very low concentrations of benzene produced from engine exhaust, tobacco smoke, or transport of petroleum products [40,41]. The relationship between long-term exposure to benzene and its associated diseases (acute myeloid leukemia, chronic myelocytic leukemia, chronic lymphocytic leukemia, acute lymphocytic leukemia, Hodgkin’s disease, etc. [42,43,44]) is relatively well known. However, it remains unclear whether any of the substances produced during metabolism of benzene exhibit the toxicity of benzene through certain mechanisms. One possible mechanism is oxidative damage caused by semiquinone and active oxygen materials produced from hydroquinone and catechol metabolites, which are products of metabolic process [45,46]. A study found that fasting blood glucose, blood insulin, and insulin resistance index were significantly increased as the urinary MuA increased. It also revealed that insulin resistance can be associated with urinary MuA alone with significant explanatory power [47]. The oxidative damage mechanism is indicated as one of the major mechanisms affecting the progression of diabetes due to increase in insulin resistance [48]. Increased insulin resistance is the important factor for the development of not only diabetes but also MetS [49]. 

We found an association between urinary MEHHP, which is a DEHP metabolite, and MetS before and after adjustment of covariates. Phthalates are used as plasticizers in various products such as plastic food packing and cosmetic products (lotions, fragrances, etc.). Therefore, they can leach into food and water leading to exposure through ingestion and dermal contact. Consequently, phthalate metabolites are detected in urine samples ubiquitously. Some studies [19,28,29] analyzed the association between phthalates metabolites and MetS. In a previous study conducted among a small number of adolescents (data from 918 people), a significant association was found between MnBP and MetS [28]. Another study revealed that higher concentrations of DEHP metabolites (the sum of MEHP, MEHHP, and MEOHP) were associated with an increased OR of MetS in men [29]. In this study, we only revealed the association between sum of DEHP metabolites and development of MetS. However, we analyzed the association between MetS and MEHP, MEHHP, and MEOHP and found that only MEHHP had a significant relationship with MetS. Because the chemical properties and health effects of each DEHP metabolite are not well known, the association between MEHHP and MetS is difficult to explain. 

In our study, blood levels of mercury and lead and urine levels of cadmium were not significantly associated with MetS after adjusting for demographic characteristics. However, some preceding studies claimed that heavy metals and MetS are interrelated, but their results were inconsistent and the possible mechanisms were not clearly explained. A study that used the 2005–2010 KNHANES data suggested that cadmium levels in urine was associated with MetS, but the result was significant only in men [16]. The covariates in this study were age, residence area, education level, smoking and drinking status, exercise, AST, and ALT. Another study showed the association between level of lead in blood and MetS using data from 1405 people [15]. These two studies used the same data, but the results were different because different covariates were adjusted. The data used in these studies (KNHANES) and those of our research (KNEHS) were considered as representative data, but the detailed variables were different. For example, KNHANES directly used blood glucose levels and blood pressure, but our study replaced it with medication use due to the absence of these variables. Because the above study revealed significant results in men, we performed another analysis using only the data of male participants. The results of our analysis were same as those reported in previous studies but not considered significant. In other previous studies, AST and ALT levels were the adjusted covariates. Although AST and ALT levels were different in both groups in our study, they were not affected by multiple logistic regression analysis. Hence, we did not use these covariates in our analysis. 

BPA and MetS were had a significant inverse relationship. Our additional analysis showed that participants with higher BPA levels had lower diabetic morbidity. One review article [50] found that BPA increases insulin resistance with the consequent risk of diabetes. However, another study [19] directly examined the relationship between type 2 diabetes and BPA and showed similar results (not significant). Hence, the relationship between BPA and MetS must be investigated further. 

Smoking was not an independent risk factor and had no significant association with MetS. This result was the same as those of previous studies [15,16]. However, some studies have found that smoking and MetS were related [10,37]. Drinking status showed a significant inverse relationship with the incidence of MetS, which might be explained by a previous study on the relationship between diabetes (component of MetS) and drinking status [51]. This study found that drinking increased insulin sensitivity and reduced the risk of diabetes. Another explanation of this inverse relationship could be reduction of alcohol consumption after diagnosing diseases comprising MetS. However, the composition of past-drinker and non-drinker in the MetS and non-MetS group did not show any clue for this possible explanation. There were large changes in the ORs and statistical significances from model 1 to model 2 by adjusting the social demographic variables. ORs of cadmium, MuA, PGA, MaA, and phthalate metabolites were all increased significantly in model 1 as the quartiles increased. However, after adjusting the social demographic variables, only MuA and MEHHP were found to have a significant relationship with MetS. It is widely known that age, gender, and educational level [52] are related with MetS, and our data also showed the same results. 

This study had some limitations. First, among the five MetS components, blood pressure, fasting glucose level, and waist circumference data were not obtained. Hence, we replaced it with current BP medication use, antidiabetic medication use, and obesity categorized based on the participants’ BMI. We could not include other MetS components, such as hyperlipidemia [53] and abdominal obesity [54]. Our operational definition was stricter than the original MetS definition, which might increase the specificity of the outcome. Second, occupational and working conditions that affect the concentrations of hazardous materials were not available and could not be used as a confounder. Third, this study was a cross-sectional study; hence, it could not show a causal relationship. 

Despite these limitations, our study results are meaningful. Although previous studies had focused the relationship between MetS and heavy metals [15,16], we expanded hazardous materials to VOC and phthalate metabolites. We revealed significant novel associations between urine levels of MuA and MEHHP concentrations and MetS after adjusting several confounders in a large number of adult participants.

## 5. Conclusions

Urine levels of MuA and MEHHP concentration were significantly associated with MetS. Because of the rarity of this study, which investigated the relationship between each VOC and phthalate metabolite with MetS, and the strict definition of the indirect measure of MetS components, further research is needed.

## Figures and Tables

**Table 1 ijerph-16-00671-t001:** Demographic and clinical characteristics of the study subjects by metabolic syndrome (MetS).

Variables Data	*N* (%)		*p*-Value
	No MetS (*n* = 4673)	MetS (*n* = 578)	
Male	2191 (46.9)	241 (41.7)	0.018 *
Female	2482 (53.1)	337 (58.3)	
Age (arithmetic mean ± SE, year)	49.87 ± 0.22	61.59 ± 0.50	<0.001 *
*Age group (year)*			
20–29	439 (9.4)	9 (1.6)	<0.001 *
30–39	860 (18.4)	23 (4.0)	
40–49	974 (20.8)	50 (8.7)	
50–59	1038 (22.2)	133 (23.0)	
60–69	855 (18.3)	197 (34.1)	
70+	507 (10.8)	166 (28.7)	
Body Mass Index (BMI) (arithmetic mean ± SE, kg/m^2^)	24.05 ± 0.05	27.02 ± 0.16	<0.001 *
*Obesity*			
Normal (BMI < 30)	4516 (96.6)	436 (75.4)	<0.001 *
Obese (BMI ≥ 30)	157 (3.4)	142 (24.6)	
*Smoking Status*			
Non-smoker	2950 (63.1)	384 (66.4)	0.100
Past-smoker	815 (17.4)	103 (17.8)	
Current smoker	908 (19.4)	91 (15.7)	
*Drinking Status*			
Non-drinker	1468 (31.4)	270 (46.7)	<0.001 *
Past-drinker	293 (6.3)	57 (9.9)	
Current drinker	2912 (62.3)	251 (43.4)	
*Education level*			
<High school	1419 (30.4)	346 (59.9)	<0.001 *
High school	1464 (31.3)	152 (26.3)	
College and more	1790 (38.3)	80 (13.8)	
*Regular exercise*			
Yes	1697 (36.3)	211 (36.5)	0.929
No	2976 (63.7)	367 (63.5)	
*Marital status*			
Single	529 (11.3)	13 (2.2)	<0.001 *
Married	3740 (80.0)	453 (78.4)	
Divorce/Separation	404 (8.6)	112 (19.4)	
*Income level* ^1^			
Good	41 (0.9)	7 (1.2)	<0.001 *
Average	3424 (73.3)	329 (56.9)	
Bad	1208 (25.9)	242 (41.9)	
Alanine aminotransferase (arithmetic mean ± SE, U/L)	23.8 ± 0.3	28.0 ± 0.8	<0.001 *
Aspartate aminotransferase (arithmetic mean ± SE, U/L)	25.1 ± 0.2	26.8 ± 0.5	<0.001 *

Note: T-test for continuous variables and chi-square test for categorical variables. ^1^ Criterion for categorizing income level: The respondents directly select among the three in the survey. * *p*-value < 0.05; SE: standard error.

**Table 2 ijerph-16-00671-t002:** Blood and urine hazardous material concentrations by metabolic syndrome (MetS) status.

Variables	No MetS (*n* = 4673)	MetS (*n* = 578)	*p*-Value
Urinary heavy metal (geometric mean ± SE)			
Urinary cadmium (μg/dL)	−0.591 ± 0.688	−0.320 ± 0.635	<0.001 *
Blood heavy metal (geometric mean ± SE)			
Blood lead (μg/dL)	0.713 ± 0.482	0.759 ± 0.487	0.031 *
Blood mercury (μg/dL)	1.180 ±0.640	1.165 ± 0.664	0.594
Urinary VOCs metabolite (geometric mean ± SE)			
Hippuric acid (g/dL)	−1.610 ± 0.016	−1.560 ± 0.493	0.321
Muconic acid (μg/dL)	4.400 ± 0.114	4.479 ± 0.032	0.021 *
Phenylglyoxylic acid (mg/dL)	−1.508 ± 0.123	−1.380 ± 0.331	0.001 *
Mandelic acid (mg/dL)	−1.563 ± 0.011	−1.455 ± 0.029	0.001 *
Sum of urinary MHA isomer (geometric mean ± SE)	−1.183 ± 0.014	−1.260 ± 0.039	0.077
Urinary phthalate metabolite (geometric mean ± SE)			
MEHHP (μg/dL)	3.275 ± 0.010	3.485 ± 0.026	<0.001 *
MEOHP (μg/dL)	2.912 ± 0.010	3.129 ± 0.029	<0.001 *
MECPP (μg/dL)	3.380 ± 0.009	3.570 ± 0.026	<0.001 *
MnBP (μg/dL)	3.570 ± 0.010	3.700 ± 0.031	<0.001 *
MBzP (μg/dL)	1.388 ± 0.015	1.552 ± 0.040	<0.001 *
Urinary bisphenol A (geometric mean ± SE) (μg/dL)	0.038 ± 0.016	0.225 ± 0.056	0.001 *

Note: T-test for all variables. Figures were log-transformed after creatinine adjustment. * *p*-value < 0.05. VOCs: volatile organic compounds; MEHHP: mono (2-ethyl-5-hydroxyhexyl) phthalate; MEOHP: mono (2-ethyl-5-oxohexyl) phthalate; MECPP: mono (2-ethyl-5-carboxypentyl) phthalate; MnBP: mono-n-butyl phthalate; MBzP: mono-benzyl phthalate.

**Table 3 ijerph-16-00671-t003:** Metabolic syndrome (MetS) risk (Odds Ratio (OR)) based on hazardous materials.

Outcome Variable	Model 1 ^1^		Model 2 ^2^		Model 3 ^3^	
	OR (95% CI)	*p*-Value	OR (95% CI)	*p*-Value	OR (95% CI)	*p*-Value
Metabolic syndrome						
Urinary cadmium						
Quartile 1	1 (Reference)		1 (Reference)		1 (Reference)	
Quartile 2	1.475 (1.102–1.973)	0.009 *	0.908 (0.668–1.233)	0.536	0.914 (0.670–1.246)	0.569
Quartile 3	1.967 (1.489–2.600)	<0.001 *	0.930 (0.689–1.255)	0.633	0.929 (0.685–1.259)	0.633
Quartile 4	3.016 (2.314–3.931)	<0.001 *	1.098 (0.815–1.479)	0.539	1.094 (0.809–1.480)	0.558
Blood lead						
Quartile 1	1 (Reference)		1 (Reference)		1 (Reference)	
Quartile 2	1.159 (0.899–1.494)	0.256	0.951 (0.728–1.244)	0.716	0.941 (0.717–1.236)	0.663
Quartile 3	1.358 (1.059–1.742)	0.016 *	0.974 (0.745–1.272)	0.844	0.999 (0.763–1.309)	0.994
Quartile 4	1.262 (0.982–1.623)	0.069	0.794 (0.601–1.049)	0.105	0.859 (0.648–1.138)	0.289
Blood mercury						
Quartile 1	1 (Reference)		1 (Reference)		1 (Reference)	
Quartile 2	0.776 (0.549–1.097)	0.152	0.883 (0.614–1.270)	0.503	0.844 (0.584–1.218)	0.364
Quartile 3	0.838 (0.604–1.161)	0.288	1.019 (0.722–1.439)	0.913	0.948 (0.669–1.344)	0.765
Quartile 4	0.863 (0.656–1.136)	0.295	1.078 (0.801–1.449)	0.621	0.990 (0.733–1.337)	0.947
Urinary muconic acid						
Quartile 1	1 (Reference)		1 (Reference)		1 (Reference)	
Quartile 2	1.133 (0.880–1.460)	0.333	1.074 (0.826–1.398)	0.593	1.066 (0.817–1.390)	0.636
Quartile 3	1.298 (1.013–1.662)	0.039 *	1.330 (1.028–1.721)	0.030 *	1.311 (1.011–1.700)	0.041 *
Quartile 4	1.234 (0.962–1.584)	0.098	1.453 (1.118–1.888)	0.005 *	1.393 (1.069–1.816)	0.014 *
Urinary phenylgloxylic acid						
Quartile 1	1 (Reference)		1 (Reference)		1 (Reference)	
Quartile 2	1.096 (0.842–1.426)	0.498	1.133 (0.861–1.490)	0.373	1.074 (0.814–1.417)	0.615
Quartile 3	1.469 (1.144–1.887)	0.003 *	1.298 (1.000–1.684)	0.050 *	1.236 (0.950–1.609)	0.115
Quartile 4	1.510 (1.177–1.938)	0.001 *	1.204 (0.930–1.560)	0.159	1.171 (0.902–1.520)	0.235
Urinary mandelic acid						
Quartile 1	1 (Reference)		1 (Reference)		1 (Reference)	
Quartile 2	0.960 (0.737–1.249)	0.759	0.862 (0.656–1.134)	0.290	0.819 (0.621–1.081)	0.158
Quartile 3	1.303 (1.016–1.672)	0.037 *	1.084 (0.836–1.406)	0.544	1.022 (0.785–1.329)	0.874
Quartile 4	1.481 (1.160–1.889)	0.002 *	1.131 (0.878–1.458)	0.341	1.097 (0.849–1.418)	0.479
Urinary MEHHP						
Quartile 1	1 (Reference)		1 (Reference)		1 (Reference)	
Quartile 2	1.746 (1.317–2.315)	<0.001 *	1.326 (0.989–1.778)	0.059	1.345 (1.001–1.808)	0.050 *
Quartile 3	1.862 (1.408–2.462)	<0.001 *	1.163 (0.866–1.561)	0.316	1.151 (0.854–1.550)	0.356
Quartile 4	2.709 (2.075–3.537)	<0.001 *	1.339 (1.003–1.788)	0.048 *	1.334 (0.996–1.787)	0.054
Urinary MEOHP						
Quartile 1	1 (Reference)		1 (Reference)		1 (Reference)	
Quartile 2	1.632 (1.231–2.164)	0.001 *	1.185 (0.883–1.589)	0.258	1.193 (0.886–1.607)	0.245
Quartile 3	1.966 (1.494–2.587)	<0.001 *	1.171 (0.875–1.567)	0.289	1.213 (0.902–1.631)	0.201
Quartile 4	2.534 (1.942–3.305)	<0.001 *	1.156 (0.862–1.550)	0.333	1.181 (0.877–1.591)	0.273
Urinary MECCP						
Quartile 1	1 (Reference)		1 (Reference)		1 (Reference)	
Quartile 2	1.691 (1.277–2.238)	<0.001 *	1.260 (0.940–1.688)	0.122	1.280 (0.952–1.721)	0.102
Quartile 3	1.870 (1.418–2.465)	<0.001 *	1.132 (0.844–1.518)	0.407	1.122 (0.834–1.509)	0.447
Quartile 4	2.579 (1.978–3.362)	<0.001 *	1.204 (0.898–1.616)	0.215	1.176 (0.873–1.584)	0.285
Urinary MnBP						
Quartile 1	1 (Reference)		1 (Reference)		1 (Reference)	
Quartile 2	1.190 (0.918–1.542)	0.189	1.020 (0.778–1.337)	0.886	1.052 (0.800–1.384)	0.716
Quartile 3	1.377 (1.069–1.774)	0.013 *	0.992 (0.759–1.297)	0.953	1.046 (0.798–1.373)	0.744
Quartile 4	1.490 (1.160–1.912)	0.002 *	0.860 (0.658–1.124)	0.268	0.897 (0.684–1.176)	0.430
Urinary MBzP						
Quartile 1	1 (Reference)		1 (Reference)		1 (Reference)	
Quartile 2	1.318 (1.015–1.711)	0.038 *	1.219 (0.930–1.598)	0.152	1.247 (0.948–1.640)	0.114
Quartile 3	1.453 (1.124–1.878)	0.004 *	1.097 (0.839–1.434)	0.5	1.100 (0.839–1.444)	0.490
Quartile 4	1.615 (1.254–2.078)	<0.001 *	1.070 (0.821–1.396)	0.616	1.090 (0.834–1.426)	0.527

* *p-*value < 0.05; ^1^ Model 1 is shown as OR and 95% CI adjusted for creatinine (excluding blood lead and mercury). ^2^ Model 2 is shown as OR and 95% CI further adjusted for age, sex, education, income, marital status. ^3^ Model 3 is shown as OR and 95% CI further adjusted for aspartate aminotransferase, alanine aminotransferase. CI: confidence intervals.

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
