# Peer review of "Association between Heavy Metals, Bisphenol A, Volatile Organic Compounds and Phthalates and Metabolic Syndrome"

_ijerph, 2019, doi:10.3390/ijerph16040671_

Round 1

Reviewer 1 Report

The authors have done a good job objectively stating their questions and methodology used to address these questions. The results found are important variables to consider in terms of contributing factors in the MetS field. While the study is unique in terms of the variables they consider including VOC and phthalate metabolites, it does however raise a little concern that the variables used by the authors to define MetS are slightly different from previously published studies. The authors do a good job about being objective and forthcoming with their findings and do state the need for further research into this molecules in relation to MetS.

Author Response

Point 1: While the study is unique in terms of the variables they consider including VOC and phthalate metabolites, it does however raise a little concern that the variables used by the authors to define MetS are slightly different from previously published studies. The authors do a good job about being objective and forthcoming with their findings and do state the need for further research into these molecules in relation to MetS.

Response 1: We appreciate the valuable comment from you. Our response to your comment is listed below.

We already described this issue in line 119 ~ 122 such as “As BP, waist circumference, and fasting glucose levels were not measured in the KNEHS, we used an operational definition for MetS based on the following criteria: current BP medication use, obesity (body mass index (BMI) ≥30) [37], and current anti-diabetic medication use.” Also we pointed out the limitation from this issue in line 251 ~ 256 such as “First, among the five MetS components, blood pressure, fasting glucose level, and waist circumference were not obtained in the data. Hence, we replaced it with current BP medication use, antidiabetic medication use, and obesity categorized based on the participants’ BMI. We could not include other MetS components, such as hyperlipidemia [55] and abdominal obesity [56]. Our operational definition was stricter than the original MetS definition, which might increase the specificity of the outcome.” This issue was caused by original data, it cannot be overcome by statistical method. Further research using close data to get conceptual MetS definition needs.

Reviewer 2 Report

The research article entitled “Association between Heavy Metals, Bisphenol A, Volatile Organic Compounds, and Phthalates and Metabolic Syndrome” emphasizes on effect of heavy metals on metabolic syndrome.

1)      The article section Abstract needs formatting in accordance with journal format

2)      Why not waist/hip ratio used instead of BMI to define Metabolic syndrome

3)      Referencing should be in accordance with journal format. Please use endnote.

4)      On what basis does authors define BMI < 30 as normal

Please consider revising the manuscript in light of comments.

Author Response

We appreciate the valuable comment from you. Our response to your comment is listed below.

Point 1: The article section Abstract needs formatting in accordance with journal format

Response 1: We modified Abstract in accordance with journal format.

Point 2: Why not waist/hip ratio used instead of BMI to define Metabolic syndrome

Response 2: Our Korean National Environmental Health Survey II (2012-2014) data does not include waist circumference and hip circumference variables. So the waist/hip ratio was replaced by BMI, another indicator of obesity. We pointed out the limitation from this issue in line 251 ~ 256 such as “……waist circumference were not obtained in the data. Hence, we replaced it with ……, and obesity categorized based on the participants’ BMI. We could not include other MetS components, such as hyperlipidemia [55] and abdominal obesity [56]. Our operational definition was stricter than the original MetS definition, which might increase the specificity of the outcome.” This issue was caused by original data, it cannot be overcome by statistical method. Further research using close data to get conceptual MetS definition needs.

Point 3: Referencing should be in accordance with journal format. Please use endnote.

Response 3: We modified the reference in accordance with journal format using the endnote.

Point 4: On what basis does authors define BMI < 30 as normal

Response 4: We classified BMI>=30 as ‘obesity’ according to the criteria of WHO (Reference [37]) as in previous studies.